# Alzheimer’s Disease Mouse as a Model of Testis Degeneration

**DOI:** 10.3390/ijms21165726

**Published:** 2020-08-10

**Authors:** Vince Szegeczki, Gabriella Horváth, Helga Perényi, Andrea Tamás, Zsolt Radák, Dóra Ábrahám, Róza Zákány, Dora Reglodi, Tamás Juhász

**Affiliations:** 1Department of Anatomy, Histology and Embryology, Faculty of Medicine, University of Debrecen, Nagyerdei krt. 98, H-4032 Debrecen, Hungary; szegeczki.vince@anat.med.unideb.hu (V.S.); helga.perenyi@gmail.com (H.P.); roza@anat.med.unideb.hu (R.Z.); 2Department of Anatomy, PTE-MTA PACAP Research Team, University of Pécs Medical School, Szigeti út 12, H-7624 Pécs, Hungary; gabriella.horvath@aok.pte.hu (G.H.); andreatamassz@gmail.com (A.T.); dora.reglodi@aok.pte.hu (D.R.); 3Research Institute of Sport Science, University of Physical Education, Budapest, Alkotas út 44, H-1123 Budapest, Hungary; radak@tf.hu (Z.R.); dora.abraham87@gmail.com (D.Á.)

**Keywords:** testis degeneration, Sox9, Alzheimer’s disease, collagen type IV, physical activity

## Abstract

Pituitary adenylate cyclase activating polypeptide (PACAP) is a neuropeptide with protective functions in the central nervous system and various peripheral organs. PACAP has the highest expression level in the testes, among the peripheral organs, and has a positive regulative role in spermatogenesis and in sperm motility. In the present study, we explored testicular degenerative alterations in a mouse model of Alzheimer’s disease (AD) (B6C3-Tg(APPswe,PSEN1dE9)85Dbo/J) and demonstrated changes in PACAP-regulated signaling pathways. In addition, the effects of increased physical activity of AD (trained AD (TAD)) mice on testis were also followed. Reduced cell number and decreased thickness of basement membrane were detected in AD samples. These changes were compensated by physical activity. Expression of PACAP receptors and canonical signaling elements such as PKA, P-PKA, PP2A significantly decreased in AD mice, and altered Sox transcription factor expression was also detected. Via this signaling mechanism, physical activity compensated the negative effects of AD on the expression of type IV collagen. Our findings suggest that the testes of AD mice can be a good model of testis degeneration. Moreover, it can be an appropriate organ to follow the effects of various interventions such as physical activity on tissue regeneration and signaling alterations.

## 1. Introduction

Pituitary adenylate cyclase activating polypeptide (PACAP) is a 38 amino acid neuropeptide, which was detected first in the central nervous system (CNS) and subsequent studies have shown PACAP expression also in various peripheral organs [1]. G-protein-coupled receptors are activated by the neuropeptide such as PAC1, VPAC1 and VPAC2. The PAC1 receptor binds PACAP with the highest affinity, while VPAC1 and VPAC2 receptors have lower binding affinity [1]. Binding of the neuropeptide to its receptors results in the transcription of various genes through the cAMP/protein kinase A (PKA) signaling pathway. Additionally, several other signaling connections and crosstalks have been identified [2,3,4,5,6], which show the complex regulatory role of PACAP in signal transduction. Testicular activation of PKA leads to the phosphorylation of Sox9 and 10 (SRY-related HMG-box) transcription factors which regulate the expression of testatin, collagen type IV and IX. Collagen type IV is one of the major components of basement membranes and its expression is partly under the control of Sox transcription factors. Activity of the Sox9 transcription factor is controlled by reversible phosphorylation by protein phosphatase 2A (PP2A) [7].

PACAP has well-defined functions in several developmental processes such as cartilage [8] and bone differentiation [4]. Among peripheral organs, the highest peripheral concentration was detected in the testis [9]. PACAP has been shown to regulate proper testis formation and spermatogenesis [7], and it has a positive effect on sperm motility [10]. As suggested from the high testicular expression, it plays a role also in testicular hormone production [11] and in testicular immune cell regulation [12]. Based on these functions, it is not surprising that lack of PACAP leads to several morphological and functional changes in PACAP knockout (KO) mice [7]. In addition, PACAP has a general cytoprotective effect against different harmful stimuli [13] and plays a role in tumor formation [14], including testicular cancer [9]. Furthermore, PACAP has a protective effect in aging [7,15], amyloidosis [16] and in formation of neurodegenerative diseases [17].

Alzheimer’s disease (AD) is the most common cause of dementia in elderly humans [18]. On the other hand, not only is the CNS involved in the manifestation of the illness, but several other disorders are associated with it, such as diabetes [19], vascular abnormalities [20] and inflammation [21]. AD is investigated mainly in the CNS, but it is shown to be a systemic disease with several pathological alterations in the periphery [21]. Overproduction and accumulation of beta amyloid peptide (Aβ)—formed from the larger amyloid precursor protein (APP)—plays an important role in the pathogenesis of this neurodegenerative disease [22]. APP is also produced in peripheral tissues and it is detectable in the testes [23]. It is presumed that APP has a role in the proper formation of cell–cell and cell–matrix interactions in the male gonad [24]. While amyloidogenesis is a well-controlled process in healthy tissues, pathologic amyloid plaques can be formed in AD [25], leading to degeneration in certain organs [26]. For the detailed analysis of AD, several animal models have been constructed [27,28], in which mutation of tau or Aβ, but most frequently double mutants, were analyzed [29,30]. These mice show signs of the disease and some AD-related degenerative changes can be identified also in the periphery [29]. The brain of B6C3-Tg(APPswe, PSEN1dE9)85Dbo/J transgenic mice [31] has been investigated in detail, but the systemic findings have not been thoroughly mapped yet. In addition to the cognitive decline, motoric dysfunction develops, and locomotion of patients decreases [32]. Controlled training postponed the manifestation of AD in human patients [33] and it has been demonstrated that continuous physical exercise affects the concentration of Aβ in plasma [33]. Various studies have been published in the last few years about the advantages of physical activity on AD formation [34] but its peripheral effects have not been discussed in detail, although increased locomotion and training in B6C3-Tg(APPswe, PSEN1dE9)85Dbo/J transgenic mice have yielded promising results to slow down AD formation [35,36]. Therefore, physical activity can be a possible intervention with potential systemic effects and positive consequences on different tissue regeneration in AD [35].

In AD, correlating with accumulating amounts of Aβ, expression of PACAP decreases [18]. On the other hand, PACAP has an important protective effect in the neurodegenerative disease [17]. We give evidence that testes of AD mice show signs of degeneration in correlation with decreased PACAP signaling activity. Moreover, we studied the effects of regular exercise on the testis in this animal model of AD, since it has been shown that exercise affects testicular function [37]. Therefore, our main goal was to find a good in vivo testis degeneration model that can show the signs of altered signaling pathways, as well as morphological and spermatogenic disorders. We propose that AD testis can be a good candidate to investigate altered spermatogenesis in vivo. Moreover, the effects of different interventions such as elevated physical activity can be easily followed and extrapolated on testis degeneration processes.

## 2. Results

### 2.1. Decreased Cell Density and Thinned Basement Membrane of Convoluted Seminiferous Tubules in Alzheimer’s Disease

Hematoxylin-eosin (HE) staining was performed to visualize morphological differences in the testis of different experimental groups. A normal phenotype of convoluted seminiferous tubules was identified with normal cellular layers of spermatogenesis in wild-type (WT) animals (Figure 1A). The number of different cells such as spermatogonia, spermatocytes and spermatozoa decreased in AD testes compared with WT (Figure 1B). Leydig cell number was also determined and reduced in AD (Figure 1B). In trained AD (TAD) animals, histological properties of convoluted seminiferous tubules partly recovered (Figure 1A) and the number of spermatogonia, spermatocytes and spermatozoa was increased (Figure 1B). Interstitial Leydig cell number also elevated after long-time training (Figure 1B). In AD animals, thickness of basement membrane was significantly thinner than in WT mice (Figure 1C). Thickness of the basement membrane showed an increase in TAD mice compared with AD mice but did not reach the normal thickness measured in WT animals (Figure 1C).

### 2.2. Decreased PAC1 Receptor Expression in Testis of AD Mice

First, RT-PCR was performed to follow the mRNA expression of PACAP receptors in total testis lysate. Previously, we have demonstrated the function of PACAP signaling in the testes of WT and PACAP KO mice [7] and we demonstrated the receptor expression in Black6 mice testes (Figure 2A). In AD mice, the mRNA expression was hardly detectable (Figure 2A), but expression of the PAC1 receptor reached the expression level of WT mice in TAD animals (Figure 2A). Protein expression level of PAC1 receptor, in correlation with the mRNA expression, diminished in AD testis (Figure 2B), but increased in the TAD experimental group (Figure 2B). VPAC1 receptor mRNA and protein expression were shown in all experimental groups with an elevation in protein expression in TAD mice compared with AD mice (Figure 2A,B). Low mRNA expression of VPAC2 was detectable in all experimental groups (Figure 2A). On the contrary, decreased VPAC2 protein expression was shown in TAD testis (Figure 2B). 

### 2.3. Reduction in PACAP Signaling Elements in AD Testis

Examination of the canonical downstream signaling pathway of PAC1 receptor showed alterations in AD animals. mRNA and protein expression of PKA decreased in AD animals compared to WT mice (Figure 3A,B), while in TAD animals, mRNA and protein expression reached the level of expression detected in WT animals (Figure 3A,B). The more active, phosphorylated form of PKA diminished in AD testis but was augmented by physical activity in TAD mice testis (Figure 3B). Interestingly, similar alterations were detected in case of PP2A (Figure 3A,B). Significant elevation of Sox10 protein expression was shown in TAD testis, while it was almost undetectable in WT and AD mice (Figure 3B). We were able to detect Sox9 mRNA and its protein in all experimental groups (Figure 3A), moreover, its more active phosphorylated form increased in TAD testis but did not alter in AD animals (Figure 3B). P-Sox9 was also revealed with immunohistochemistry in convoluted seminiferous tubules and it showed a strong immunoreactivity around cell nuclei (Figure 3C). P-Sox9 positive cells were visible randomly distributed in seminiferous tubules of WT animals and only the peripheral cells showed immunoreactivity in AD testis (Figure 3C). In TAD samples various cell types with peripheral and central location started to re-express P-Sox9 transcription factor (Figure 3C). In AD testis, expression of P-Sox9 was detectable without accumulation around the nuclei of cells of different stages of spermatogenesis (Figure 3C). Elevated immunoreactivity was visible in TAD testis and signals became stronger around the cell nuclei (Figure 3C).

### 2.4. Expression of Basement Membrane Components Altered in Seminiferous Tubules in AD

Expression of protein components of the basement membrane in seminiferous tubules was also investigated. mRNA expression of collagen type IV and testatin was detected in all experimental groups (Figure 4A). mRNA of collagen type IX was hardly detectable in WT and AD testis but showed an increased mRNA expression in TAD animals (Figure 4A). Protein expression of collagen type IV decreased in AD group and increased in TAD testis (Figure 4B). Interestingly, protein expression of collagen type IX was not in correlation with mRNA expression. It was reduced both in AD and TAD testis (Figure 4B). Protein expression of testatin elevated in AD and also in TAD animals (Figure 4B). Collagen type IV was examined with immunohistochemistry in seminiferous tubules. It was detectable in the basement membrane of convoluted seminiferous tubules in WT testis (Figure 4C). On the other hand, collagen type IV immunoreactivity was hardly detectable in basement membrane of AD testis (Figure 4C). Due to active movement, the immunoreactivity of this molecule was elevated in TAD testis and showed an accumulation in the basement membrane (Figure 4C).

## 3. Discussion

AD is a chronic neurodegenerative disorder and its incurability demands a growing socioeconomic pressure to investigate the detailed mechanism of the disease formation [38]. Although the modified signaling pathways have been widely investigated in the CNS, alterations in the periphery have not been discussed in detail. AD results in a systemic disorder in humans and in animal models with various organs showing pathological signs [39,40]. Degeneration in AD has been demonstrated in testis, kidneys and pancreas [41,42] indicating that Aβ accumulation can have deteriorating effects also in the periphery. 

PACAP has been identified in the hypothalamo–hypophyseal system [1] and its protective role has been shown in several pathological CNS disorders, such as ischemia [43], spinal cord injury [44], inflammation [45], peripheral nerve regeneration [46] and retinopathy [47]. PACAP has been shown to ameliorate morphological and behavioral signs of Parkinson’s disease [48] and has a protective role in AD [49]. PACAP also plays a protective role in aging, proven by the accelerated aging in various tissues of KO mice [15,50] but not in testis where delayed aging of testicular cells was demonstrated [51]. As PACAP is needed for the proper development and appropriate function of several tissues, such as bone [4], cartilage [8] and testis [7], it is not surprising that its lack disturbs the formation of long bones [4] and callus formation [52]. Furthermore, the reduced expression of PACAP alters the histological structure of the testis, modifies the steps of spermatogenesis, negatively affects sperm morphology and motility [7,10]. The lack of the neuropeptide also disturbs the Sox transcription factor function and alters the proper basement membrane formation, leading to degradation-like changes in the testis [7]. PACAP receptor distribution and specific expression in spermatogonia, spermatocytes and spermatozoa have been demonstrated [9], moreover, PACAP receptor independent uptake has also been shown [53]. Subsequently, PACAP regulation seems to be an essential signaling pathway, which maintains the normal structure of testis and proper spermatogenesis [7]. Therefore, three experimental groups were set up: a WT (wild type) to demonstrate the potential function of PACAP signaling in testis, an AD transgenic mice to investigate the systemic effect of the disease and a trained AD (TAD) mouse population to determine the systemic function of physical activity and its connection with PACAP regulation in the testis. As we demonstrated, the AD-induced systemic effects resulted in a cell number reduction in convoluted seminiferous tubules and reduced Leydig cell number, furthermore, in decreased thickness of their basement membrane. These findings further strengthen the hypothesis that AD triggers a degeneration process of peripheral organs. It has been published that increased physical activity has a preventive and attenuating effect in AD [35]. Sex hormone reduction has been published in AD which further strengthens the systemic effect of the degenerative illness [54]. Moreover, it is also known that PACAP signaling is involved in the regulation of mechanotransduction [55]. However, it has not yet been investigated whether PACAP has a systemic effect in increased mechanical activity or only locally modifies the activation of various signaling cascades. We found that physical activity normalized the thickness of basement membrane and elevated the reduced cell numbers in convoluted seminiferous tubules and interstitium. Subsequently, in this present study we attempted to provide evidence that the testis of AD transgenic mice is a good model where the systemic effects of a degenerative disease and those of increased physical activity can be followed. 

Among the PACAP binding receptors, PAC1 receptor has the highest expression in the testicle [56]. All PACAP binding receptors could be detected in WT mice [7,9]. Expression of the high affinity binding PAC1 receptor [1] decreased in the testis of AD animals, similar to the CNS [18]. Physical exercise can be protective against Aβ neurotoxicity [34]. In our study, the expression of PAC1 receptor was nearly normalized in TAD animals. VPAC receptors’ expression has been shown in AD [57], but their alterations in the periphery have not been investigated yet. In AD testes, VPAC expression was detectable but their expression did not show a consistent alteration suggesting an AD independent expression. 

Most of the studied downstream signaling pathways of PAC1 receptor activation are related to elevated activation of adenylate cyclase enzyme [1]. Subsequently, the increased cAMP concentration activates PKA [58] which has a higher activity after being phosphorylated (P-PKA) [7]. In testis of AD mice, expression of PKA and P-PKA decreased, similar to that in neuronal tissues in AD [59]. Increased physical exercises can modulate the activity of PKA and subsequently the expression of some target genes [60]. We demonstrated that training could compensate this decline in AD mice. In male gonads, P-PKA phosphorylates transcription factor Sox9 [7]. Sox9 and Sox10 are redundant factors in the SoxE family [61], also demonstrated in testes of PACAP KO mice [7]. In testes of AD mice Sox9 expression was elevated. Similar to our results, elevated expression of Sox9 has been demonstrated in astrocytes of AD [62]. On the other hand, no alterations were detected in the quantity of P-Sox9 and no signal of Sox10 appeared in testis of AD, but the expression of these transcription factors increased in TAD animals. P-Sox9 immunoreactivity was very low in AD mice, while signal intensity was stronger around nuclei of WT and TAD mice. Sox9 expression has been published in Sertoli cells to regulate testis development and it can be translocated to cells of spermatogenesis via blood–testis barrier and regulates their differentiation process [63]. In AD testes only, peripheral cells expressed Sox9 where Sertoli cells and spermatogonia are located proposing a blood–testis barrier disorder. Mechanical load has been demonstrated to elevate the expression and phosphorylation of Sox9 and PACAP also altered its expression in testis [7] and in cartilage [8]. These results also suggest that physical activity triggers multiple signaling cascades which normalize differentiation processes in spermatogenesis partly via Sox9 activation. In our previous research we have shown that PP2A expression and activity have been elevated in PACAP KO mice [7]. The activity and expression of this Ser/Thr phosphatase are reduced in AD [64], and its inhibition augmented Aβ accumulation in the CNS [65]. Although PP2A activity has been reduced in direct mechanical load [66], no data can be found about its systemic function during elevated physical activity. We demonstrated that the expression of PP2A was reduced in AD but elevated in TAD testis parallel with the appearance of strong Sox9 and 10 expressions. These results suggest a PP2A independent activation of Sox9 [67] and a compensatory effect of Sox10 to maintain the proper spermatogenesis. Elevated PP2A expression may have a preventive role in AD via inhibition of the hyperphosphorylation of certain factors such as tau [68] substituting or normalizing PP2B activity [69]. 

In testis, Sox9 enhances the expression of collagen type IV, collagen type IX and testatin, components of basal membrane in seminiferous tubules [70]. These components play a crucial role in formation and signaling modulation of the blood–testis barrier [71]. In neuronal tissues of AD patients reduced expression of collagen type IV was detected [72] and collagen type IV accumulation inhibited Aβ formation [73]. Subsequently, the functions of neurovascular basement membranes are altered in AD [74] which may have an effect on blood–brain barrier. In PACAP KO mice, collagen type IV expression, which diminished and altered morphology of spermatids, was detected [7], implying a blood–testis barrier alteration in PACAP KO animals. The decreased expression of collagen type IV in testes of AD mice was also confirmed by immunohistochemistry. These results propose an Aβ-induced degradation in testes and possible disturbance of spermatogenesis in the result of altered blood–testis barrier functions. Surprisingly, training increased the expression of collagen type IV and testatin further strengthening the positive effect of physical activity on AD. This may also result in maintaining normal basal membrane integrity. Diminished collagen type IX expression in TAD animals highlights a physical activity independent regulation.

Finally, we can conclude that investigation of effects of toxins on testicular tissues and cell types mainly followed in in vitro experiments [75] where the complex process and alterations in testis dysfunction is not analyzable. Testicular torsion is a serious urological disease leading to testicular damage and it is used as an in vivo degeneration model [76]. Certain degradation processes can be followed in these models, but these interventions may induce inflammation or hypoxia which can have false effects on testis biology. Testis of transgenic AD mice showed pathological signaling alterations in PACAP regulated pathways and spermatogenic disorders as well. Additionally, the effect of a systemic intervention was well analyzable without any other biological modulation during in vivo circumstances.

## 4. Material and Methods

### 4.1. Animals 

Generation of male Alzheimer-transgenic (B6C3-Tg(APPswe,PSEN1dE9)85Dbo/J) mice were used in the experiments. Three-month-old wild type (WT) (no transgenic modulation without any training) (*n* = 5), Alzheimer transgenic mice (AD) (*n* = 5) and trained Alzheimer disease (TAD) mice (*n* = 5) were kept under light/dark cycles of 12/12 h with free access to food and water. Alzheimer transgenic mice were trained on treadmill four times per week for one hour divided into 10 sessions. One session contained 2 min low intensity running (10 m/min) and 4 min high intensity running (20 m/min). The study was carried out in accordance with ethical guidelines (ethical permission number for this study: PEI/001/2105-6/2014 (09.07.2014), Semmelweis University, Hungary). Genotyping was performed using Phire Animal Tissue Direct PCR Kit (Thermo Fischer Scientific, Waltham, MA, USA) according to the manufacturer’s instructions.

### 4.2. Light Microscopical Morphology

Testes of 3-month-old mice were washed in PBS three times and fixed in a 4:1 mixture of absolute ethanol and 40% formaldehyde, then embedded in paraffin. Serial sections were made, and Hematoxylin–Eosin staining was performed (HE, Sigma-Aldrich, MO, USA). Staining protocol was carried out according to the instructions of manufacturer. Photomicrographs were taken using a DP74 camera (Olympus Corporation, Tokyo, Japan) on Olympus B×53 microscope (Olympus Corporation, Tokyo, Japan). Seminiferous tubules were investigated according to the histological localization and morphological characteristics of different cell types of spermatogenesis. Spermatogonia, spermatocytes and spermatozoa were counted in cross sections of convoluted seminiferous tubules [77]. For identification of spermatogonia round, darkly stained cells were counted at the level of basement membrane. Without identifying type I and type II spermatocytes specifically, cells with round nuclei were counted close to the lumen of convoluted seminiferous tubules. In the luminal surface of convoluted seminiferous tubules, cells with dark small elongated nuclei were counted as spermatozoa. Eosinophil cells were determined as Leydig cell in the interstitium. Cell numbers were determined in cross sections of 10 separated convoluted seminiferous tubules by 3 operators in 5 independent experiments. Leydig cells were counted in 10× magnification photomicrographs by 3 operators in 5 independent experiments. For the measurement of basement membrane thickness, ImageJ 1.40 g freeware was used. We drew a perpendicular line in the basement membrane and the pixels were determined and converted to micrometer. Twenty independent measurements were made in one slide and 5 independent testes were used per experimental group. 

### 4.3. Immunohistochemistry

Immunohistochemistry was performed on wild type (WT), AD mice, and TAD mice testis samples to visualize localization of P-Sox9 and collagen type IV (Col. IV). Testes were fixed in a 4:1 mixture of absolute ethanol and 40% formaldehyde and washed in 70% ethanol. After embedding serial sections were made, deparaffination was then followed by rinsing in PBS (pH 7.4). Non-specific binding sites were blocked with PBS supplemented with 1% bovine serum albumin (Amresco LLC, Solon, OH, USA), then samples were incubated with polyclonal P-Sox9 (Sigma-Aldrich, MO, USA), or Col. IV (Abcam, Cambridge, UK) antibodies at a dilution of 1:600 at 4 °C overnight. For visualization of the primary antibodies, anti-rabbit Alexa fluor 555 secondary antibody (Life Technologies Corporation, Carlsbad, CA, USA) was used at a dilution of 1:1000. Samples were mounted in Vectashield mounting medium (Vector Laboratories, Peterborough, England) containing 4’,6-diamidino-2-phenylindol (DAPI) for nuclear DNA staining. For negative controls, anti-rabbit Alexa fluor 555 was used without the primary antibodies. Photomicrographs were taken using DP74 camera (Olympus Corporation, Tokyo, Japan) on Olympus B×53 microscope (Olympus Corporation, Tokyo, Japan). Images were acquired using cellSense Entry 1.5 software (Olympus, Shinjuku, Tokyo, Japan) with constant camera settings to allow a comparison of fluorescent signal intensities. Images of Alexa555 and DAPI were overlaid using Adobe Photoshop version 10.0 software. Contrast of images were equally increased without changing constant settings.

### 4.4. RT-PCR Analysis

Testes of WT, AD mice and TAD mice were mechanically ground and were dissolved in Trizol (Applied Biosystems, Foster City, CA, USA), after 30 min incubation on 4 °C total RNA was isolated. RNA was harvested in RNase-free water and stored at −70 °C. Reverse transcription was performed by using High Capacity RT kit (Applied Biosystems, Foster City, CA, USA). For the sequences of primer pairs and details of polymerase chain reactions, see Table 1. Primers were designed with PrimerBlast freeware and obtained from IDT (Integrated DNA Technologies, Interleuvenlaan, Belgium). Amplifications were performed in a thermal cycler (Labnet MultiGene™ 96-well Gradient Thermal Cycler; Labnet International, Edison, NJ, USA) as follows: 95 °C, 2 min, followed by 35 cycles (denaturation, 94 °C, 30 s; annealing for 45 s at optimized temperatures as given in Table 1; extension, 72 °C, 90 s) and then 72 °C, 7 min. Actin was used as internal control. PCR products were analyzed using a 1.2% agarose gel containing ethidium bromide. 

### 4.5. Western Blot Analysis

Testes of WT, AD and TAD mice were washed in physiological saline and stored at −70 °C. Samples were mechanically disintegrated with a tissue grinder in liquid nitrogen. Then they were collected in 100 μL of homogenization RIPA (radioimmunoprecipitation assay)-buffer (150 mM sodium chloride; 1.0% NP40, 0.5% sodium deoxycholate; 50 mM Tris, pH 8.0) containing protease inhibitors (Aprotinin (10 μg/mL), 5 mM Benzamidine, Leupeptin (10 μg/mL), Trypsine inhibitor (10 μg/mL), 1 mM PMSF, 5 mM EDTA, 1 mM EGTA, 8 mM Na-Fluoride, 1 mM Na-orthovanadate). The suspensions were sonicated by pulsing burst for 30 s at 40 A (Cole-Parmer, Illinois, USA). Total cell lysates for Western blot analyses were prepared. In total, 40 μg protein was separated in 7.5% SDS–polyacrylamide gels for the detection of PAC1, VPAC1, VPAC2, PKA, Sox9, P-Sox9, Sox10, PP2A, collagen type IV (Col. IV), collagen type IX (Col. IX), testatin and actin. Proteins were transferred electrophoretically to nitrocellulose membranes and exposed to the primary antibodies overnight at 4 °C in the dilution as given in Table 2. After washing for 30 min with phosphate-buffered salin with 0.1% Tween 20 (PBST), membranes were incubated with the peroxidase-conjugated secondary antibody anti-rabbit IgG in a 1:1500 (Bio-Rad Laboratories, CA, USA) or anti-mouse IgG in 1:1500 (Bio-Rad Laboratories, CA, USA) dilution. Signals were detected with enhanced chemiluminescence (Advansta Inc., Menlo Park, CA, USA) according to the instructions of the manufacturer. Actin was used as an internal control. Signals were developed with gel documentary system (Fluorchem E, ProteinSimple, CA, USA). 

### 4.6. Statistical Analysis

All data are representative of at least five independent experiments. For all figures, the samples of the same WT, AD and TAD animals were chosen with their inner control for the better comparison. Only one demonstrative photo from the same animal group was used in every figure, although the changes were based on the 5 results. Statistical analysis was performed by one-way analysis of variance (ANOVA), followed by Tukey’s HSD (honestly significant difference) post-hoc test (SigmaPlot, Systat Software, Inc., San Jose, CA, USA). Threshold for statistically significant differences as compared to control samples was set at * *p* < 0.05 and to AD samples # *p* <0.05.

## 5. Conclusions

Our results suggest that the testis of AD mice can be a good candidate for investigation of spermatogenic disorders or testis degeneration processes. There are many structural and functional similarities between the different cells of CNS and testis such as Sertoli—germ cell and neuron-glial cell relation is comparable and shows similar basic signaling activation mechanisms [78]. Furthermore, correct microcirculation and barrier formation show similar alterations both in testis and CNS [12,79,80]. Thus, it can be presumed that examination of signaling pathways in AD testis could be a plausible way for modeling pathologic degeneration conditions of testis where the effects of various interventions such as physical activity can be investigated.

## Figures and Tables

**Figure 1 ijms-21-05726-f001:**
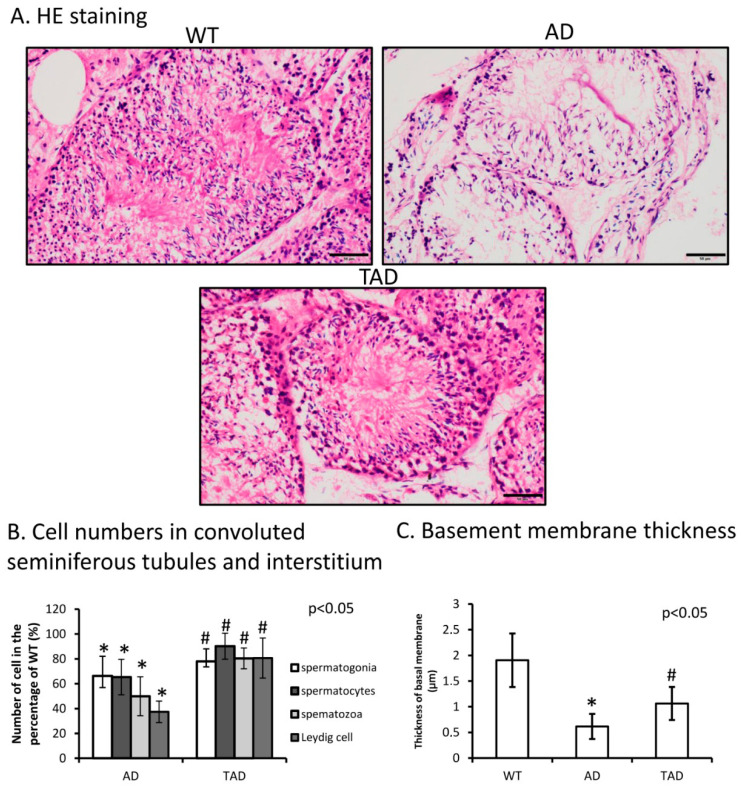
(**A**) Representative microphotograph of wild-type (WT), Alzheimer’s disease (AD) and trained AD (TAD) convoluted seminiferous tubules stained with hematoxylin-eosin (HE). Original magnification was 40×. Scale bar, 20 µm. (**B**) Cell number (spermatogonia, spermatocytes, spermatozoa) determination of convoluted seminiferous tubules and Leydig cell in the interstitium. (**C**) Thickness analysis of testicular basement membrane. Representative data of 5 independent experiments. Asterisks indicate significant (* *p* < 0.05) difference in thickness of basement membrane compared to WT and (^#^
*p* < 0.05) compared to AD.

**Figure 2 ijms-21-05726-f002:**
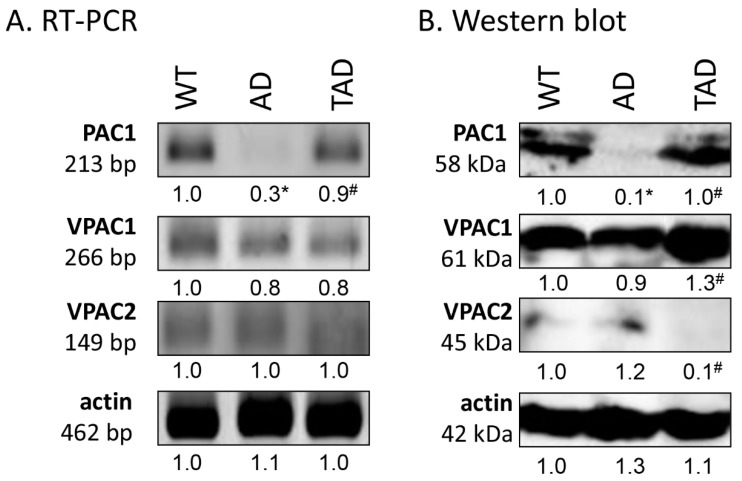
Representative photos of mRNA (**A**) and protein (**B**) expression of pituitary adenylate cyclase activating polypeptide (PACAP) receptors in the testis. Optical density of signals was measured, and results were normalized to the optical density of WT. For panels (**A**,**B**), numbers below the signals represent integrated densities of signals determined by ImageJ software. Asterisks indicate significant (* *p* <0.05) alteration of expression as compared to the WT and (^#^
*p* <0.05) compared to AD. Representative data of 5 independent experiments. For RT-PCR reactions and for Western blot, actin was used as controls.

**Figure 3 ijms-21-05726-f003:**
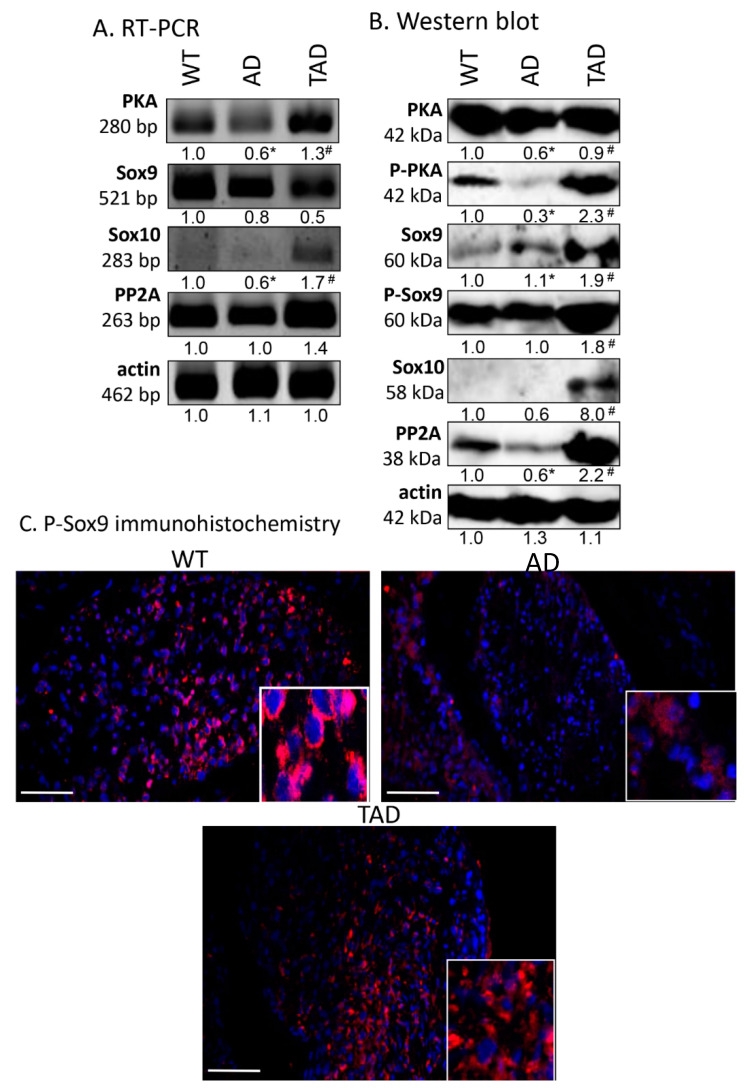
mRNA (**A**) and protein (**B**) expression of PACAP signaling in testis. For panels (**A**,**B**), numbers below the signals represent integrated densities of signals determined by ImageJ software. Asterisks indicate significant (* *p* <0.05) alteration of expression as compared to the WT and (^#^
*p* <0.05) compared to AD. Representative data of 5 independent experiments. For RT-PCR reactions and for Western blot, actin was used as controls. (**C**) Immunohistochemistry of P-Sox9 in seminiferous tubules. Magnification was made with 40× objective. Scale bar: 20 µm.

**Figure 4 ijms-21-05726-f004:**
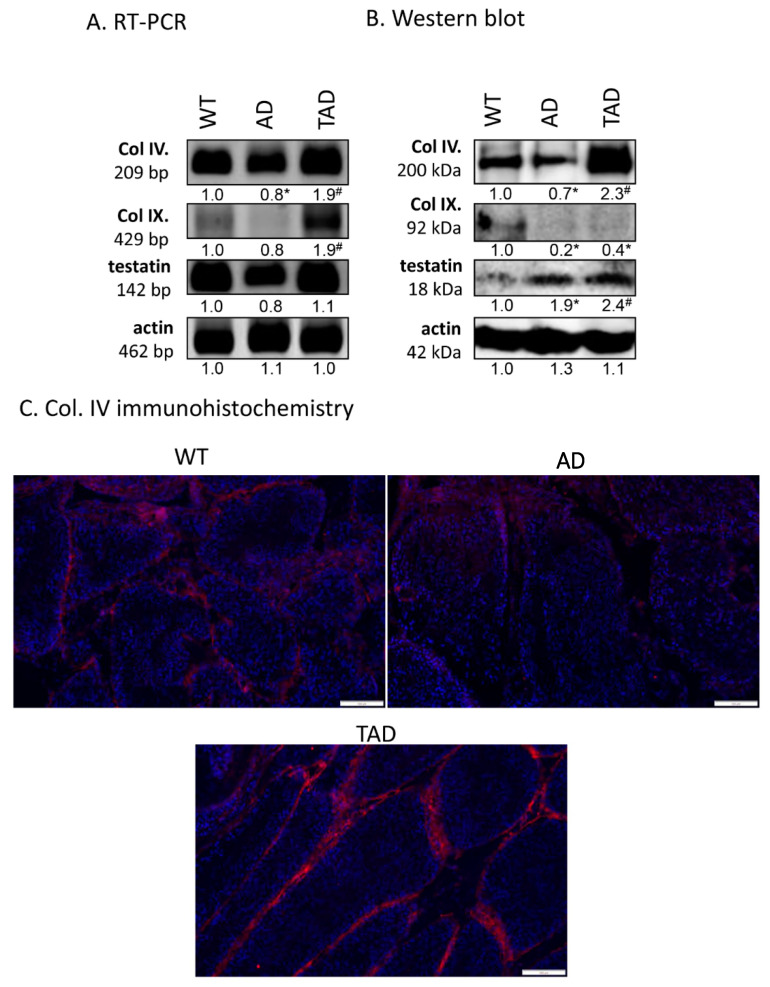
mRNA (**A**) and protein (**B**) expression of Col. IV, Col. IX and testatin. For RT-PCR and for Western blot reactions, actin was used as control. Optical density of signals was measured, and results were normalized to the optical density of controls. For panels (**A**,**B**), numbers below the signals represent integrated densities of signals determined by ImageJ software. Asterisks indicate significant (* *p* < 0.05) alteration of expression as compared to the WT and (^#^
*p* <0.05) compared to AD. Representative data of 5 independent experiments. For RT-PCR reactions and for Western blot, actin was used as controls. (**C**) Immunohistochemistry of Col. IV in seminiferous tubules. Magnification was made with 20× objective. Scale bar: 50 µm.

**Table 1 ijms-21-05726-t001:** Nucleotide sequences, amplification sites, GenBank accession numbers, amplimer sizes and PCR reaction conditions for each primer pair are shown.

Gene	Primer	Nucleotide Sequence (5′→3′)	GenBank ID	Annealing Temperature	Amplimer Size (bp)
PAC1	sense	TATTACTACCTGTCGGTGAAG (912–932)	NM_007407.4	52 °C	213
	antisense	ATGACTGCTGTCCTGCTC (1107–1124)			
VPAC1	sense	TTT GAG GAT TTC GGG TGC (974–991)	NM_011703.4	53 °C	266
	antisense	TGG GCC TTA AAG TTG TCG (1222–1239)			
VPAC2	sense	CTC CTG GTA GCC ATC CTT (805–822)	NM_009511.2	53 °C	149
	antisense	ATG CTG TGG TCG TTT GTG (936–953)			
PKA (Prkaca)	sense	GCAAAGGCTACAACAAGGC (847–865)	NM_008854	53 °C	280
	antisense	ATGGCAATCCAGTCAATCG (1109–1126)			
Sox9	sense	GTA CCC GCA TCT GCA CAA CG (378–397)	NM_011448	62 °C	521
	antisense	GTG GCA AGT ATT GGT CAA ACT CAT T (874–898)			
Sox10	sense	ACG ACT GGA CGC TGG TGC (535–552)	NM_011437.1	58 °C	283
	antisense	CGC CGA GGT TGG TAC TTG TAG (797–817)			
PP2A (Ppp2ca)	sense	CTC TGC GAG AAG GCT AAA (288–305)	NM_017039.2	54 °C	436
	antisense	TGA TTC CCT CGG AGT ATG (706–723)			
collagen IV (Col4a1)	sense	TCG GCT ATT CCT TCG TGA TG (4963–4982)	NM_009931.2	52 °C	209
	antisense	GGATGGCGTGGGCTTCTT (5154–5171)			
collagen IX (Col9a3)	sense	CAG GTT CCG ATG GTC TTC C (1357–1375)	NM_009936.2	55 °C	492
	antisense	CTG TTG CTC CCT TGT CCC (1831–1848)			
testatin (Cys9)	sense	CTG GAG GGA GAA GGT AAA (274–291)	NM_009979.1	51 °C	142
	antisense	CAG GCA GGT GAA GGT ATT (398–415)			
actin (Actb)	sense	GCCAACCGTGAAAAGATGA (419–437)	NM_007393.5	54 °C	462
	antisense	CAAGAAGGAAGGCTGGAAAA (861–880)			

**Table 2 ijms-21-05726-t002:** Tables of antibodies used in the experiments.

Antibody	Host Animal	Dilution	Distributor
Anti-Col. IX.	rabbit, polyclonal,	1:500	Abcam, Cambridge, UK
Anti-Col. IV.	rabbit, polyclonal,	1:800	Abcam, Cambridge, UK
Anti-Testatin	rabbit, polyclonal,	1:300	Santa Cruz Biotechnology, Dallas, TX, USA
Anti-PKA C	rabbit, polyclonal,	1:600	Cell Signaling, Danvers, MA, USA
Anti-Sox9	rabbit, polyclonal,	1:600	Abcam, Cambridge, UK
Anti-P-Sox9	rabbit, polyclonal,	1:800	Sigma-Aldrich, St. Louis, MO, USA
Anti-Sox10	rabbit, polyclonal,	1:500	Abcam, Cambridge, UK
Anti-PP2A C	rabbit, polyclonal,	1:600	Cell Signaling, Danvers, MA, USA
Anti-Actin	mouse, monoclonal,	1:10,000	Sigma-Aldrich, St. Louis, MO, USA

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
