# Peer review of "Alzheimer’s Disease Mouse as a Model of Testis Degeneration"

_ijms, 2020, doi:10.3390/ijms21165726_

Round 1

Reviewer 1 Report

The manuscript titled: „Alzheimer’s disease mouse as a model of testis degeneration” presents valuable results. In immunohistochemical study and gene and protein analysis, Authors evidenced that the testis of Alzheimer’s disease can be a good model of testis degeneration. Generally, that manuscript is interesting but it requires a major improvement:

  • As Authors written in Line 99, Fig. 2 has already been published in another paper of Authors (reference No 14) – it is unacceptable to publish it once again because it is autoplagiarism.
  • Introduction and Discussion require re-organisation. Apart from information about PACAP, in Introduction, Authors should place the information about AD and the role of physical activity in AD. Also, about signaling pathways which are studied (Sox9 and 10, PKA, PP2A, collagen IV, IX, testatin) – any connections?
  • Last paragraph of the Introduction should clearly point out the purpose of the study and the scheme of experimental procedure (now it is in Discussion in lines 184-186).
  • In Discussion Authors should point out other models of testis degeneration and discuss novel model in comparison with previous ones. What is the advantage/disadvantage of that model?
  • 4 – Authors use two abbreviations for “collagen” – Col 4 and Col IV - which is correct?
  • Abbreviations, such as Sox, Coll, PKA, PP2A – should be explained in the text, when they are mentioned for the first time.
  • Line 253 – “correct microcirculation”
  • Statistical analysis - for three experimental groups, the one-way ANOVA and post-hoc test, not Student’s test, is recommended.

Author Response

  1. As Authors written in Line 99, Fig. 2 has already been published in another paper of Authors (reference No 14) – it is unacceptable to publish it once again because it is autoplagiarism.

We apologize for the misunderstanding and we rephrased the sentence in the “Results” chapter. We agree with the Reviewer that we have already published data about the function of PACAP signaling in testis of PACAP KO mice and WT littermates but expression of PACAP receptors was not published in that article. On the other hand, in these experiments Black6 mouse was used as a WT control where the expression of PACAP receptors has not been investigated yet. It was likely that we can demonstrate the receptor expression in the WT mice but it was needed to show it to compare with AD and TAD testes where PACAP receptor expression was not investigated yet. Therefore, we referred the article in the discussion. Moreover,  Prisco et al. recently published an investigation about the distribution and location of PACAP receptors in testis.

  1. Introduction and Discussion require re-organisation. Apart from information about PACAP, in Introduction, Authors should place the information about AD and the role of physical activity in AD. Also, about signaling pathways which are studied (Sox9 and 10, PKA, PP2A, collagen IV, IX, testatin) – any connections?

We agree with the suggestions of the Reviewer and additional sentences are given in the Introduction.

  1. Last paragraph of the Introduction should clearly point out the purpose of the study and the scheme of experimental procedure (now it is in Discussion in lines 184-186).

The main goal of the research is pointed out in the end of the Introduction.

  1. In Discussion Authors should point out other models of testis degeneration and discuss novel model in comparison with previous ones. What is the advantage/disadvantage of that model?

The comparison is inserted in the Discussion chapter.

  1. 4 – Authors use two abbreviations for “collagen” – Col 4 and Col IV - which is correct?

Corrected to Col. IV.

  1. Abbreviations, such as Sox, Coll, PKA, PP2A – should be explained in the text, when they are mentioned for the first time.

Abbreviations are explained.

  1. Line 253 – “correct microcirculation”

Corrected

  1. Statistical analysis - for three experimental groups, the one-way ANOVA and post-hoc test, not Student’s test, is recommended.

We have corrected our statistical analyses and inserted in the text and modified in the figures.

Reviewer 2 Report

In their manuscript Authors provide insights to consider testis of Alzheimer’s disease mice as a model of testis degeneration, moreover they show changes in PACAP-regulated signaling pathways. The manuscript is well written, and goals are clear.

However, the manuscript suffers of major revision before publishing on Ijms.

Below there are some useful comments to improve the manuscript.

In general fluorescence images should be improved both in low and high magnification

Fig 1 Scale bar is not evident.

It is not clear why the thickness is expressed in pixels and not in microns.

Higher magnification is needed for each experimental group. It seems that the AD testis shows a whole

structural disorganization that Authors should describe, analysing interstitial Leydig cells too. In literature it has been described that “One of the most common observations associated with AD onset is decreased levels of sex hormones, including estrogens, progesterone and androgens"  (Sex Hormones and Alzheimer’s Disease” by Wafik Said Bahnasy, Yasser A. El-Heneedy and Ehab A. El-Seidy, December 24th 2017

DOI: 10.5772/intechopen.72561)

Furthermore, AD mice “ show a beneficial action of androgens on cognition, pathology, and biochemical measures" (Dubal, D.B., Broestl, L. & Worden, K. Sex and gonadal hormones in mouse models of Alzheimer’s disease: what is relevant to the human condition?. Biol Sex Differ 3, 24 (2012). https://doi.org/10.1186/2042-6410-3-24).

It should be very interesting to evaluate the number of AD Leydig cells in each experimental class.

Fig 3 A, B

It is not clear, if values were normalized on the controls, why actin has values higher than 1.

Please explain in detail the method

Fig 3 C

It should be better to individuate the cytotypes that express P-Sox9. This is more important because the distribution of the immunoreactivity is clearly wide for WT, in periphery in AD and throughout the testis zonation in TAD. Authors should discuss these differences in P-Sox9 distribution.

Page 4 L129 This is not mandatory but “immunopositivity” is not as useful as “immunoreactivity”

P 6 L 135 It is unclear why if 5 independent experiments have been carried out the actin band are the same for all figures.

P6 L137 Actin is not the best control for PCR.

Truly there is not perfect control for PCR but GAPDH or nuclear proteins should be preferred.

Fig 4B  L155 Controls are saturated hence they are not useful at all for optical densitometry

P10 L269 The protocol to number the cell types in testes is based on the object count of 2D images taken with no specific, systematic, and random procedure. This old-fashioned way is strongly biased by operator-based and procedure-based errors, leading to a virtual no viable and useful quantitative information. It is conversely useful for qualitative description exclusively.

Neither increasing cross sectioning nor the independent experiments increase the usefulness of these quantitative measurements.

Authors should use other appropriate quantitative analysis on 2D images to become confident about the number of cytotypes in the testes; they need strong standardization method and accurate procedure for sampling, sectioning, and analysis.

P10 L 275

“According to the histological localization and characteristic of different cell types of spermatogenesis such as spermatogonia, spermatocytes and spermatozoa were counted in cross sections of convoluted seminiferous tubules.”

This sentence seems to lack the subject

P10 L283 If the method is explained in literature please provide reference. However, the procedure suffers of previous described concerns (P10 L269)

P10 L297 the omission of the primary antibody is not enough to consider negative a control. Pre-adsorption should be performed.

P10 L304 In general, PCR is not useful for quantitative measurements due to internal technical limitations, thus being useful only for the presence/absence of the targeted RNA.

Hence trying to perform quantitative analysis of RNA expression nowadays is a simply non-sense.

qPCR must be used (however taking into account its technical limitations) to carry out quantitative analysis.

P11 L308 How the primer pairs were obtained?

Please provide detailed information

P11 L 314 The evaluation of optical densities on WB or PCR band should be performed very accurately.

The integrated density method implemented in ImageJ needs a calibration procedure that accurately assign values to black/white pixels. There is no evidence that this procedure was carried out.

The calibration should be performed on a curve obtained with the antigen/antibody signal.

P12 L335 To obtain suitable signal for optical density evaluation control (and samples) should not be saturated as they appear in figures.

P12 L338 Please improve the statistical method information by adding the software used and eventually the correction of the test used.

How many animals were used?

How many testes were analysed, and for each method how many used?

P12 L339 Authors state that 5 WT, 5 TAD, and 5 AD mice were used, hence 10 testes for experimental group.

To have "at least 5 independent experiments" as stated Authors should perform 5 times this single experimental setup. Are the Authors stating that?

Otherwise, data represents 1 experiment with 5 animals for each group.

References 92 references seem to be too much for a regular manuscript. If it is possible please consider decreasing this number

Author Response

In their manuscript Authors provide insights to consider testis of Alzheimer’s disease mice as a model of testis degeneration, moreover they show changes in PACAP-regulated signaling pathways. The manuscript is well written, and goals are clear.

However, the manuscript suffers of major revision before publishing on Ijms.

Below there are some useful comments to improve the manuscript.

In general fluorescence images should be improved both in low and high magnification

  1. Fig 1 Scale bar is not evident.

Scale bar has been thickened.

  1. It is not clear why the thickness is expressed in pixels and not in microns.

We determined the thickness of basement membrane and it is converted to microns.

  1. Higher magnification is needed for each experimental group. It seems that the AD testis shows a whole

We made higher magnification photomicrographs.

  1. Structural disorganization that Authors should describe, analysing interstitial Leydig cells too. In literature it has been described that “One of the most common observations associated with AD onset is decreased levels of sex hormones, including estrogens, progesterone and androgens"  (Sex Hormones and Alzheimer’s Disease” by Wafik Said Bahnasy, Yasser A. El-Heneedy and Ehab A. El-Seidy, December 24th 2017

DOI: 10.5772/intechopen.72561)

Furthermore, AD mice “ show a beneficial action of androgens on cognition, pathology, and biochemical measures" (Dubal, D.B., Broestl, L. & Worden, K. Sex and gonadal hormones in mouse models of Alzheimer’s disease: what is relevant to the human condition?. Biol Sex Differ 3, 24 (2012). https://doi.org/10.1186/2042-6410-3-24).

It should be very interesting to evaluate the number of AD Leydig cells in each experimental class.

Thank you for the kind suggestion of the Reviewer Leydig cell number determination can further strengthen the degeneration processes in AD testis. We have determined the number Leydig cells in the interstitium and discussed in the manuscript.

  1. Fig 3 A, B

It is not clear, if values were normalized on the controls, why actin has values higher than 1. Please explain in detail the method

 Explanation is inserted in the manuscript.

  1. Fig 3 C

It should be better to individuate the cytotypes that express P-Sox9. This is more important because the distribution of the immunoreactivity is clearly wide for WT, in periphery in AD and throughout the testis zonation in TAD. Authors should discuss these differences in P-Sox9 distribution.

 We agree with the Reviewer and P-Sox9 localization is discussed in the manuscript.

  1. Page 4 L129 This is not mandatory but “immunopositivity” is not as useful as “immunoreactivity”

We corrected it in the manuscript.

  1. P 6 L 135 It is unclear why if 5 independent experiments have been carried out the actin band are the same for all figures.

We agree with the Reviewer that not every experiment was done at the same time.  All data are one representative from at least five independent experiments. For all figures the samples of the same WT, AD and TAD animals were chosen with their inner control for the better comparison. Only one demonstrative photo from the same animal group running at the same time was used in every figure, although the changes were based on the 5 results. In Material and Methods chapter we gave a detailed discussion.

  1. P6 L137 Actin is not the best control for PCR.

Truly there is not perfect control for PCR but GAPDH or nuclear proteins should be preferred.

We agree with the Reviewer that GAPDH can be a good inner control. We have tested GAPDH in the first experiment and the expression was the same as the actin. Therefore, we used the same inner control in PCR and Western blot reactions for the better comparison. First GAPDH test is attached below. WT, AD, TAD negControl. product size 411 bp.

  1. Fig 4B  L155 Controls are saturated hence they are not useful at all for optical densitometry

 During the analysis we tried to increase the quality of the bands by using Supersignal West Femto kit (PierceTM, MA, USA) but the background of the signals became extremely strong. We have tried to improve the Western blots with better blocking or higher protein concentration without better results, therefore, we repeated the experiments 5 times and we made statistical analysis of the data obtained from these 5 independent experiments. Photomicrographs were made by Fluorchem E geldocumentary system integrated camera cooled down to -40°C which increased the sensitivity. Before optical density measurements we made individual calibrations on every photomicrographs. Then we increased the magnification of the photos to reach individual pixel size and with freehanded method we precisely circled the lanes and measured the integrated density of the area. The Reviewer has right that operator-based errors can happen during the measurement procedure, therefore, we measured the pixels in 5 independent experiments by 3 independent operators which reduce the possibility of procedure-based errors. Then the 5 independent results were compared and statistical analysis has been done. In the Figures we presented data of the experiment out of the five which best represented the average results. We agree with the reviewer that these blots could be better and saturated photos are not the best for optical analyses, but despite of our efforts to improve the quality of those, we did not succeed in that.

  1. P10 L269 The protocol to number the cell types in testes is based on the object count of 2D images taken with no specific, systematic, and random procedure. This old-fashioned way is strongly biased by operator-based and procedure-based errors, leading to a virtual no viable and useful quantitative information. It is conversely useful for qualitative description exclusively.

Neither increasing cross sectioning nor the independent experiments increase the usefulness of these quantitative measurements.

Authors should use other appropriate quantitative analysis on 2D images to become confident about the number of cytotypes in the testes; they need strong standardization method and accurate procedure for sampling, sectioning, and analysis.

For standardization of sectioning first we placed the testis in a 4:1 mixture of absolute ethanol and 40% formaldehyde solution. After 16 hours hardened, fixed testes were halved according to their transverse axis and embedded in paraffin in a standardized way. After staining 3 operators independently counted cells in 5 independent experiments which reduced the errors in cell determination procedure.  We agree with the Reviewer that these are rather semiquantitative data but the tendency of the alterations supports our main findings that AD results a degeneration process in testis.   

  1. P10 L 275

“According to the histological localization and characteristic of different cell types of spermatogenesis such as spermatogonia, spermatocytes and spermatozoa were counted in cross sections of convoluted seminiferous tubules.”

This sentence seems to lack the subject

Sentence is rephrased in the manuscript.

  1. P10 L283 If the method is explained in literature please provide reference. However, the procedure suffers of previous described concerns (P10 L269)

Reference is inserted in the manuscript.

  1. P10 L297 the omission of the primary antibody is not enough to consider negative a control. Pre-adsorption should be performed.

We tested the PAC1R antibody in PAC1R KO mouse previously. We have already tested several antibodies such as Sox9, PP2A, PKA, P-PKA in cartilage but negative controls are not available for these antibodies. We tried to use siRNA for PP2A and PKA on cell lines but silencing of these proteins resulted an apoptotic process of these cell lines (Juhasz et al. 2014). For VPAC1 and VPAC2 antibodies we used blocking peptides for negative controls.

Juhász T, Matta C, Katona É, Somogyi C, Takács R, Gergely P, Csernoch L, Panyi G, Tóth G, Reglődi D, Tamás A, Zákány R. Pituitary adenylate cyclase activating polypeptide (PACAP) signalling exerts chondrogenesis promoting and protecting effects: implication of calcineurin as a downstream target.PLoS One. 2014 Mar 18;9(3):e91541.

  1. P10 L304 In general, PCR is not useful for quantitative measurements due to internal technical limitations, thus being useful only for the presence/absence of the targeted RNA.

Hence trying to perform quantitative analysis of RNA expression nowadays is a simply non-sense.

qPCR must be used (however taking into account its technical limitations) to carry out quantitative analysis.

We completely agree with the Reviewer the Q-PCR would be an appropriate method for quantitative analysis. We tried to isolate pure RNA for Q-PCR reactions, but the complexity of samples was so small and during grinding mechanism we probably lost part of the mRNA, which did not let us run acceptable Q-PCR reactions. Therefore, we made 5 independent experiments to analyze the results from at least 5 independent animals. We used only one demonstrative photo from the same animal in every figure, although the changes were based on the 5 results. However, we completely agree with the reviewer that band saturation can affect the results. On the other hand, the tendency of the results show and support the signaling alterations in testicular degeneration process of AD.

  1. P11 L308 How the primer pairs were obtained?

Please provide detailed information

Thank you for the suggestion, detailed information is inserted in the manuscript.

  1. P11 L 314 The evaluation of optical densities on WB or PCR band should be performed very accurately.

The integrated density method implemented in ImageJ needs a calibration procedure that accurately assign values to black/white pixels. There is no evidence that this procedure was carried out.

The calibration should be performed on a curve obtained with the antigen/antibody signal.

We apologize for not giving a detailed explanation in Material and methods chapter. We tried to give better explanation and inserted in the manuscript.  Furthermore, we made graphs about the result for better presentation of densitometry.  

  1. P12 L335 To obtain suitable signal for optical density evaluation control (and samples) should not be saturated as they appear in figures.

See our answers in question number 10 (Fig.4 Lane 155). Graphs about the statistical analysis is added to the manuscript for better understanding the optical density results.

  1. P12 L338 Please improve the statistical method information by adding the software used and eventually the correction of the test used.

Software is given in the manuscript. (SigmaPlot, Systat Software, Inc., San Jose, CA, USA)

  1. How many animals were used?

How many testes were analysed, and for each method how many used?

P12 L339 Authors state that 5 WT, 5 TAD, and 5 AD mice were used, hence 10 testes for experimental group.

To have "at least 5 independent experiments" as stated Authors should perform 5 times this single experimental setup. Are the Authors stating that?

Otherwise, data represents 1 experiment with 5 animals for each group.

In the experiments 5 animals were used from every experimental group (WT, AD, TAD). One testis from the same animal was halved and used for Western blot and PCR-reaction. The other testis was placed into fixative and then precisely halved according to its transverse axis. One half was used for histological staining and the other half was used for immunohistochemical analysis. 

  1. References 92 references seem to be too much for a regular manuscript. If it is possible please consider decreasing this number

Although we decreased the number of the references in the manuscript but during the revision several other paper had to be referred which did not let really reduce the number of the references.

Round 2

Reviewer 1 Report

The manuscript titled: „Alzheimer’s disease mouse as a model of testis degeneration” is now much better. All my recommendations are corrected.

I noticed a mistake in Fig.3 and 4.: point C. “Denistometric analysis” should be corrected to “Densitometric analysis”

In References, position 31 and 33 – remove colon before the number of pages.

In my opinion, after these corrections that manuscript can be accepted for publication.

Author Response

We removed and corrected the typing mistakes.

Reviewer 2 Report

Authors deeply improved their manuscript.

Notwithstanding this the reviewer holds the opinion that quantitative analysis must be obtained by quantitative measurements to keep high scientific quality. The Author's response to the reviewer's comments about densitometry of qPCR and WB are not enough to explain why they kept in the revised manuscript quantitative analysis based on qualitative measurements.

Authors surely know the whole amount of bias, errors and misunderstanding that the subjectivity of observations brings to quantitative analysis.

The manuscript keep its high scientific quality without densitometric histograms based on no ratiometric analysis. Biased histograms decreases its scientific quality.

Some typos (e.g."denistometric") should be corrected.

94 references are too much. Please consider decreasing this number

Author Response

We tryed to avoid to use the "significant" expression in the Results and Discussion chapter and  to highlight the qualitative anylisis in the manuscript. We also removed the histogrmas from the figures and results. Therefore, the typing mistakes disappeared from the figures. 

We decreased the references to 80. 

Round 3

Reviewer 2 Report

Authors should erase any sentence about densitometry in the materials and methods section.

Author Response

We removed the sentences about densitometry in Material and Methods.